# Multivariate Analysis Reveals That Unsubstituted β-Ring and C8-Keto Structures Are Important Factors for Anti-Inflammatory Activity of Carotenoids

**DOI:** 10.3390/nu13113699

**Published:** 2021-10-21

**Authors:** Yuki Manabe, Nami Tomonaga, Takashi Maoka, Tatsuya Sugawara

**Affiliations:** 1Division of Applied Biosciences, Graduate School of Agriculture, Kyoto University, Kitashirakawa Oiwake-cho, Sakyo-ku, Kyoto 606-8502, Japan; tomonaga.nami.52v@kyoto-u.jp (N.T.); sugawara.tatsuya.6v@kyoto-u.ac.jp (T.S.); 2Research Institute for Production and Development, 15 Shimogamo Morimoto-cho, Sakyo-ku, Kyoto 606-0805, Japan; maoka@mbox.kyoto-inet.or.jp

**Keywords:** carotenoid, retinoid, Toll-like receptor, cytokine, non-infectious inflammation, anti-inflammatory activity, nuclear factor-kappa B, interferon regulatory factor, structure-activity relationship, multivariate analysis

## Abstract

Carotenoids are natural lipophilic pigments with substantial health benefits. Numerous studies have demonstrated the anti-inflammatory activities of carotenoids, especially toward lipopolysaccharide-induced inflammatory responses. As such, there are few reports on the evaluation and comparison of the anti-inflammatory activities of carotenoids against inflammation induced by other stimuli. In this study, we used pathogen-associated molecular patterns, proinflammatory cytokines, degenerated proteins, and chemical irritants as inflammatory inducers to evaluate the anti-inflammatory activities of eight different carotenoids. Each carotenoid showed characteristic anti-inflammatory activities; thus, we conducted a multivariate analysis to clarify the differences among them. Unsubstituted β-ring (i.e., provitamin A) and C8-keto structures of carotenoids were found to be crucial for their inhibitory effects on the activation of nuclear factor-kappa B and interferon regulatory factors, respectively. Furthermore, we found that β-carotene and echinenone treatment increased intracellular retinoid levels in monocytes and that the retinoids showed the similar activities to β-carotene and echinenone. Taken together, the intake of both provitamin A and C8-keto carotenoids (e.g., siphonaxanthin and fucoxanthin) might be effective in improving the inflammatory status of individuals. A multivariate analysis of anti-inflammatory activities is a useful method for characterizing anti-inflammatory compounds.

## 1. Introduction

Inflammation is a physiological response to external assaults, such as pathogen invasion. Although inflammation itself plays an important role in restoring homeostasis, excessive or chronic inflammation is known to damage the body and lead to various deleterious conditions such as sepsis and atherosclerosis. Multiple inflammatory stimuli and their respective receptors are involved in the body’s response to external assaults. Invading pathogens are detected by pattern recognition receptors such as Toll-like receptors (TLRs) that induce inflammatory responses, including proinflammatory cytokine production. In addition, proinflammatory cytokines, such as tumor necrosis factor (TNF)-α, interleukin (IL)-1β, and interferon (IFN)-γ, evoke inflammatory responses. In addition to pathogen-derived molecules, degenerated proteins and chemical irritants are known to induce the production of proinflammatory cytokines. In fact, a TLR4 antagonist is used for the treatment of sepsis, and advanced glycation end products (AGEs)—a group of degenerated proteins—and a receptor for AGEs (RAGE) have received attention as therapeutic targets for atherosclerosis [1].

Carotenoids are low-molecular-weight lipophilic pigments produced by photosynthetic organisms and fungi. According to the Carotenoid Database, more than 1000 carotenoids have been identified thus far [2]. Human beings acquire approximately 50 kinds of carotenoids from their diet [3,4], and many studies have demonstrated the health benefits of these carotenoids. As well as provitamin A and antioxidant activities, the anti-inflammatory activity of carotenoids has been well investigated. A meta-analysis of randomized clinical trials indicated that carotenoids caused a significant reduction in the levels of C-reactive protein, a marker of inflammation [5]. Several epidemiological studies have indicated an inverse association between serum or plasma carotenoid levels and atherosclerosis [6,7,8]. Numerous cultured cell studies have also demonstrated the anti-inflammatory effects of carotenoids; however, most of the studies used lipopolysaccharides (LPSs) to evoke inflammatory responses. We previously evaluated the effect of carotenoids on inflammatory responses induced by another stimulant, AGEs, and found for the first time that siphonaxanthin showed anti-inflammatory activity [9]. Based on those results, in this study we evaluated the anti-inflammatory activity of carotenoids using multiple inflammatory inducers.

Because of the chemical diversity of carotenoids, their structure-activity relationships have been well investigated. We previously reported that non-polar carotenoids (e.g., β-carotene) and C8-keto carotenoids (e.g., siphonaxanthin and fucoxanthin) exhibited anti-inflammatory effects via suppression of lipid raft translocation of TLR4 [10]. It has also been shown that the 7,8-acetylenic bond found in alloxanthin might be important for the inhibition of LPS-induced proinflammatory cytokine expression [11]. Moreover, the C4-keto group found in astaxanthin enhanced nitric oxide production in both LPS- and IFN-γ-stimulated cells [12]. Based on these studies, we selected eight carotenoids to evaluate and compare their anti-inflammatory activities (Figure 1).

Circulating leukocytes, especially monocytes, are considered an important source of proinflammatory cytokines [13]. Once monocytes receive inflammatory stimuli, numerous proinflammatory genes are induced mainly via the activation of nuclear factor-kappa B (NF-κB) and interferon regulatory factors (IRFs). In this study, we evaluated the anti-inflammatory effects of carotenoids using a human monocytic cell line, THP-1, and reporter gene assays for NF-κB and IRFs. In addition, since some inflammatory stimuli could affect cellular proliferation, we evaluated the 3-(4,5-di-methylthiazol-2-yl)-2,5-diphenyltetrazolium bromide (MTT) reductase activity. Finally, we attempted to determine the characteristics of the anti-inflammatory activities of each carotenoid, using multivariate analyses.

## 2. Materials and Methods

### 2.1. Cells

THP1-Dual cells (InvivoGen, San Diego, CA, USA) are commercially available transfected THP-1 (human monocytic cell line) cells expressing secreted embryonic alkaline phosphatase (SEAP) upon NF-κB activation. They also secrete Lucia luciferase, a synthetic gene designed to mimic the luciferase activity of marine copepods, under the control of IRF activation. We cultured them as recommended by the manufacturer and used passage numbers below 20.

### 2.2. Reagents

LPS from Escherichia coli 0111:B4 was obtained from Santa Cruz Biotechnology (Santa Cruz, CA, USA). Pam_3_CSK_4_ (P3C) and phorbol 12-myristate 13-acetate (PMA) were purchased from Abcam (Cambridge, UK). Polyinosinic-polycytidylic acid sodium salt (pIC), recombinant human IFN-γ, and AGN 193109 were obtained from Sigma Aldrich (St. Louis, MO, USA). Recombinant human TNF-α and IL-1β were obtained from Pepro Tech (Rocky Hill, NJ, USA). Glycated bovine serum albumin (AGE-BSA) was purchased from Bio Vision (Milpitas, CA, USA). Amyloid-β 1–42 (Aβ1–42) was purchased from Peptide Institute (Osaka, Japan). Carotenoids were prepared as previously reported [14,15,16]. In brief, alloxanthin and halocynthiaxanthin were isolated from sea squirts (*Halocynthia roretzi*), echinenone was purified from sea urchins (*Anthocidaris crassispina*), fucoxanthin was purified from brown algae (*Undaria pinnatifida*), lutein was isolated from spinach (*Spinacia oleracea*), and siphonaxanthin was purified from green algae (*Codium fragile*). Fucoxanthinol was prepared from fucoxanthin by hydrolysis with lipase as described previously [17]. Astaxanthin and zeaxanthin were purchased from AG Scientific (San Diego, CA, USA) and EXTRASYNTHESE (Lyon, France), respectively. β-Carotene, retinal, and retinoic acid were purchased from FUJIFILM Wako Pure Chemical (Osaka, Japan), and 4-keto retinal and 4-keto retinoic acid were obtained from Toronto Research Chemicals, Inc. (North York, ON, Canada). The extinction coefficient of each carotenoid was used for quantification [18]. The purity was above 95% as estimated by HPLC and TLC analyses. The other chemicals, media, and solvents used in the experiments were of commercially available reagent grade.

### 2.3. Carotenoid Treatment

Carotenoids were dispersed in the culture medium using dimethyl sulfoxide (DMSO) as a vehicle. The final DMSO concentration was 0.1%. THP1-Dual cells were incubated in carotenoid-containing medium for 24 h, followed by the addition of the indicated stimulants. Resting control cells and vehicle control cells were incubated with the vehicle-containing medium for 24 h, followed by no stimulation or indicated stimulation, respectively.

### 2.4. Reporter Gene Assay

THP1-Dual cells at a density of 2.5 × 10⁵ cells/mL in a 96-well plate were cultured with the indicated stimulants for 24 h. The concentrations of the stimulants were as follows: P3C, 100 ng/mL; pIC, 100 µg/mL; LPS, 100 ng/mL; TNF-α, 100 ng/mL; IL-1β, 10 ng/mL; IFN-γ, 100 ng/mL; AGE-BSA, 100 µg/mL; Aβ1–42, 25 µM; PMA, 100 ng/mL. To evaluate NF-κB activation, 20 µL of supernatant was mixed with 100 µL of QUANTI-Blue solution. After incubating for 60 min at 37 °C, the visible absorption at 640 nm was measured using a spectrophotometer. For the analysis of IRF activation levels, the luciferase activity in 20 µL of supernatant was measured with a luminometer 4 s after the automated injection of 50 µL of QUANTI-Luc solution.

### 2.5. MTT Assay

After stimulation, MTT was added to cell cultures at a final concentration of 1 mg/mL, followed by incubation at 37 °C for 30 min. After discarding the medium, 2-propanol (120 µL) was added to each well, and the visible absorption at 570 nm was measured using a spectrophotometer.

### 2.6. LC-MS/MS Analysis

Total lipids were extracted from the cells as described previously [19] and dissolved in 0.25 mL of methanol. After dilution with 0.75 mL of acetonitrile, 5 μL of each sample was analyzed using a Prominence HPLC system (Shimadzu, Kyoto, Japan) coupled to a QTRAP 5500 mass spectrometer (AB Sciex, Framingham, MA, USA). To quantify β-carotene, astaxanthin, siphonaxanthin, fucoxanthin, and fucoxanthinol, a Unison UK-C18 column (2.0 × 150 mm, 5 µm, Imtakt, Kyoto, Japan) was used with the following conditions: temperature, 40 °C; flow rate, 0.2 mL/min; mobile phase, 0.1% formic acid in water (A) and 0.1% formic acid in methanol/2-propanol (50/50, *v*/*v*) (B). Gradient elution was performed as follows: 0–3 min, 85% B; 3–10 min, 85%–98% B linear; 10–20 min, 98% B; 20–25 min, 98%–85% B linear; and 25–30 min, 85% B. For quantification of echinenone, zeaxanthin, alloxanthin, lutein, and retinoids, an Ascentis Express HILIC column (2.1 × 150 mm, 5 µm, Sigma Aldrich) was used with the following conditions: temperature, 40 °C; flow rate, 0.2 mL/min; mobile phase, 44 mM ammonium formate in water at pH 3.0 (adjusted using formic acid) (A) and acetonitrile containing the same volume of formic acid as in A (B). Gradient elution was performed as follows: 0–12 min, 92%–70% B linear; 12–15 min, 70%–92% B linear; and 15–40 min, 92% B. Each target was monitored in multiple reaction monitoring (MRM) mode. The MRM pairs are listed in Appendix A. The carotenoids and retinoids were quantified from standard curves and normalized to the protein content determined using the DC protein assay kit (Bio-Rad, Hercules, CA, USA).

### 2.7. Multivariate Analysis

The data from the stimulated cells were imported into SIMCA 15 (Umetrics, Umeå, Sweden) and scaled using the Pareto scaling method. A principal component analysis (PCA) and orthogonal partial least squares discriminant analysis (OPLS-DA) were conducted to clarify the differences in the anti-inflammatory profiles of each carotenoid.

### 2.8. Statistical Analysis

A one-way analysis of variance (ANOVA) with the Tukey-Kramer post-test was performed using BellCurve for Excel Ver. 7.0 (Social Survey Research Information, Tokyo, Japan). Differences were considered statistically significant at *p* < 0.05.

## 3. Results

### 3.1. Determination of the Treatment Concentration

The treatment concentration is an important experimental condition to investigate the structure-activity relationship. According to our previous studies, in many cases, siphonaxanthin exhibits more potent biological activities than the other carotenoids [9,15,20,21]. Hence, we used siphonaxanthin as a reference to determine the treatment concentration. As we reported previously [9], siphonaxanthin significantly suppressed AGEs-induced NF-κB activation at 0.5 and 1.0 µM (−21% and −29%, respectively; Appendix A). LPS- and TNF-α-induced NF-κB activation were significantly inhibited by siphonaxanthin at 1.0 µM (−19% and −16%, respectively), whereas at 0.5 µM, they were suppressed significantly but only slightly (−7% and −9%, respectively; Appendix A). Taken together, the treatment concentration of carotenoids was set at 1.0 µM in the following experiments.

### 3.2. Effects on TLR-Mediated Inflammatory Responses

The TLR1/2 heterodimer recognizes tri-acyl lipoprotein, which is one of the components of bacterial cell walls. In this study, we used P3C, a synthetic analog of the tri-acylated *N*-terminal part of bacterial lipoproteins, to activate the TLR1/2-mediated pathway. As shown in Figure 2A, P3C-induced NF-κB activation was significantly inhibited by pretreatment with β-carotene (−16%), echinenone (−13%), siphonaxanthin (−8.4%), and fucoxanthin (−7.9%), and IRF activation was significantly suppressed by siphonaxanthin (−45%) and fucoxanthin (−37%). None of the carotenoids significantly reduced MTT reductase activity (Figure 2B).

TLR3 is a receptor for dsRNA produced from infection by dsRNA viruses. We evaluated the effects of carotenoids on TLR3-mediated inflammatory responses using a synthetic analog of dsRNA, pIC. Although no carotenoid significantly suppressed pIC-induced NF-κB activation, the activation of IRFs was significantly inhibited by siphonaxanthin (−47%) and fucoxanthin (−25%; Figure 2A). These two carotenoids also significantly decreased the MTT reductase activity (−13% and −13%, respectively; Figure 2B), but the degree of inhibition of IRF activation was greater than that of MTT reductase activity. These results indicate that reduction of cellular proliferation was not responsible for the inhibitory effects on IRFs.

LPS is an endotoxin produced by Gram-negative bacteria and is known to induce inflammation by binding to TLR4. As shown in Figure 2A, LPS-induced NF-κB activation and IRF activation were significantly suppressed by β-carotene (−17% and −29%, respectively), echinenone (−13% and −35%, respectively), and siphonaxanthin (−15% and −63%, respectively), whereas fucoxanthin significantly inhibited only the activation of IRFs (−40%). However, these four carotenoids did not show any significant effect on MTT reductase activity (Figure 2B).

### 3.3. Effects on Proinflammatory Cytokine-Induced Inflammatory Responses

The TNF-α-induced activation of NF-κB and IRFs was significantly inhibited by β-carotene (−28% and −45%, respectively), echinenone (−21% and −42%, respectively), siphonaxanthin (−16% and −59%, respectively), and fucoxanthin (−4.9% and −37%, respectively; Figure 3A). Astaxanthin did not suppress TNF-α-induced NF-κB activation but significantly inhibited the activation of IRFs (−20%). Alloxanthin alone significantly decreased MTT reductase activity (−8%; Figure 3B) but did not suppress the activation of both NF-κB and IRFs.

IL-1β-induced NF-κB activation was significantly suppressed by pretreatment with β-carotene (−26%), echinenone (−21%), and siphonaxanthin (−17%), and the activation of IRFs was suppressed by siphonaxanthin (−34%; Figure 3A, middle panel). These inhibitory carotenoids did not decrease MTT reductase activity (Figure 3B, middle panel).

Unlike TNF-α and IL-1β, IFN-γ did not upregulate NF-κB activity under our experimental conditions (Figure 3A, right panel). IFN-γ-induced IRF activation was significantly lowered by β-carotene (−36%), echinenone (−33%), zeaxanthin (−45%), lutein (−48%), siphonaxanthin (−55%), and fucoxanthin (−56%; Figure 3A, right panel). Fucoxanthin significantly reduced the MTT reductase activity (−9%; Figure 3B, right panel), but the degree of reduction was lower than that of the inhibition of IRF activation.

### 3.4. Effects on Non-Infectious Inflammatory Responses

Prolonged high blood glucose levels, as seen in diabetic patients, leads to the formation of AGEs, which are known to induce inflammation in monocytes through binding with RAGE. In this study, we used BSA-derived AGEs (AGE-BSA) as a model. Siphonaxanthin significantly suppressed the AGE-BSA-induced activation of NF-κB and IRFs (−16% and −45%, respectively) without inhibiting cellular proliferation (Figure 4A,B).

Aβ1–42 is an intrinsically disordered protein consisting of 42 amino acid residues and is considered to be involved in the pathogenesis of Alzheimer’s disease. It is also known that Aβ1–42 induces inflammatory responses not only in microglia but also in monocytes by binding with some receptors [22,23]. No carotenoid significantly suppressed the upregulation of NF-κB activity in Aβ1–42-stimulated cells (Figure 4A). We could not detect the significant upregulation of IRF activity in control cells, whereas siphonaxanthin and fucoxanthin significantly decreased IRF-regulated luciferase activity (−35% and −28%, respectively). Aβ1–42 stimulation slightly but significantly decreased MTT reductase activity in resting cells, which were not affected by these two carotenoids (Figure 4B).

PMA, an ester derivative of phorbol diterpene from croton oil, is known to activate protein kinase C directly, resulting in NF-κB activation, growth arrest, and macrophage differentiation in monocytes. Similar to the results obtained for Aβ1–42, no carotenoid significantly suppressed PMA-induced NF-κB activation, and we could not detect significant upregulation of IRF activity in the control cells (Figure 4A). In contrast, β-carotene and echinenone significantly upregulated IRF transcriptional activity (+92% and +39%, respectively) compared to control cells. As shown in Figure 4B, PMA stimulation significantly decreased MTT reductase activity, which was ameliorated by pretreatment with β-carotene (+38%), echinenone (+46%), and siphonaxanthin (+13%), indicating that they suppressed PMA-induced growth arrest.

### 3.5. PCA Reveals Differences between the Anti-Inflammatory Profiles of the Carotenoids

Each carotenoid exhibited distinct characteristic effects on NF-κB, IRFs, and MTT reductase activities (i.e., distinct anti-inflammatory profiles) in stimulant-exposed THP1-Dual cells. To clarify the differences between the anti-inflammatory profiles of all the carotenoids, we conducted PCA. In the score plot (Figure 5A), zeaxanthin, astaxanthin, alloxanthin, and lutein were located close to the control, indicating that the anti-inflammatory effects of these four carotenoids were relatively weak. β-Carotene and echinenone, as well as siphonaxanthin and fucoxanthin, were located in close proximity to each other, indicating that they have similar anti-inflammatory profiles. In the loading plot (Figure 5B), NF-κB activity was located in the fourth quadrant (i.e., opposite to the position of β-carotene and echinenone in the score plot), and IRF activities were located in the first quadrant (i.e., opposite to the position of siphonaxanthin and fucoxanthin in the score plot). MTT reductase activities were located near the origin, indicating that the effects on cellular proliferation slightly contributed to the grouping.

### 3.6. Cytotoxicity and Cellular Accumulation Cannot Explain the Results of PCA

Cytotoxicity and cellular accumulation are important factors that explain the differences in biological activities. As shown in Figure 6A, carotenoid treatment did not show any remarkable effect on cell density compared to vehicle treatment. According to our LC-MS/MS analysis, each carotenoid was detected only in carotenoid-treated cells. In fucoxanthin-treated cells, its deacetylated form, fucoxanthinol, was also detected, and the ratio of fucoxanthinol to fucoxanthin was approximately 1:18. As shown in Figure 6B, alloxanthin accumulated the most in the cells, followed by fucoxanthin (including fucoxanthinol), siphonaxanthin, and astaxanthin. The cellular accumulation of β-carotene, echinenone, zeaxanthin, and lutein was much lower than that of the other carotenoids. Altogether, neither cytotoxicity nor cellular accumulation could explain the results of the PCA, suggesting that other factors such as chemical structure might be important.

### 3.7. Increase in Intracellular Retinoid Levels Is Important for the Effects of β-Carotene and Echinenone

To better define the inflammatory responses that are characteristic of β-carotene and echinenone, we conducted OPLS-DA of these two carotenoids and the other carotenoids. As shown in the S-plot (Figure 7A), suppression of the TNF-α-NF-κB axis and promotion of the PMA-IRF axis were characteristic of the effects of both β-carotene and echinenone. To explore the possible mechanisms, we first examined whether antagonism of the retinoic acid receptor (RAR) counteracts the effects of β-carotene. In the presence of AGN 193109, an antagonist for RAR, β-carotene neither suppressed the TNF-α-NF-κB axis nor promoted the PMA-IRF axis (Appendix A). Based on their chemical structures, β-carotene is metabolized to two molecules of retinal, and echinenone is metabolized to retinal and 4-oxo-retinal (Figure 7B). Retinal is further converted to retinoic acid, which in turn exerts various biological functions; therefore, we quantified retinal, retinoic acid, 4-oxo-retinal, and 4-oxo-retinoic acid using an LC-MS/MS analysis (Appendix A). As shown in Figure 7C, the β-carotene treatment significantly increased cellular levels of retinal, retinoic acid, and 4-oxo-retinal and tended to increase 4-oxo-retinoic acid levels (*p* = 0.057). The echinenone treatment significantly increased cellular retinal and 4-oxo-retinal levels. Finally, we evaluated the effects of the four aforementioned retinoids on TNF-α-induced NF-κB activation and PMA-induced IRF activation and found that the effects of the retinoids were the same as those of β-carotene and echinenone (Figure 7D,E).

### 3.8. C8-Keto Carotenoids Inhibit P3C- and LPS-Induced IRF Activation

Figure 8A shows the S-plot of OPLS-DA of the C8-keto carotenoids (i.e., siphonaxanthin and fucoxanthin) and the other carotenoids. The suppression of the P3C-IRF axis and LPS-IRF axis was characteristic of the anti-inflammatory effects of siphonaxanthin and fucoxanthin. Unfortunately, apocarotenoids which could be generated by cleaving the C8-keto carotenoids are not commercially available. Hence, we used an HPLC-photodiode array system to investigate whether apocarotenoids are present in the cells. If apocarotenoids are present, it is expected to be detected at 420 nm or 360 nm [24]. In this study, THP1-Dual cells were cultured for up to 48 h (total of 24-h pretreatment and 24-h stimulation) in a carotenoid-containing medium. No peak which could be considered as apocarotenoids was detected in the cells received 1.0 µM of siphonaxanthin for 48 h (Appendix A). We detected trace amounts of dehydro-metabolites of siphonaxanthin, but they also have a C8-keto structure (unpublished data [25]). Taken together, we considered that C8-keto carotenoid, and not the cleavage products, inhibited IRF activation. We evaluated the inhibitory effects of two other C8-keto carotenoids, halocynthiaxanthin and fucoxanthinol (Figure 8B), on the P3C-IRF and LPS-IRF axes. Similar to siphonaxanthin and fucoxanthin, halocynthiaxanthin and fucoxanthinol significantly suppressed IRF activation in P3C- and LPS-exposed THP1-Dual cells (Figure 8C,D, respectively).

## 4. Discussion

The PCA revealed that anti-inflammatory carotenoids can be divided into two groups: one consisting of provitamin A carotenoids (β-carotene and echinenone), which tend to decrease NF-κB activity, and the other consisting of C8-keto carotenoids (siphonaxanthin and fucoxanthin), which tend to decrease IRF activity. According to the Carotenoid Database, provitamin A carotenoids with a C8-keto structure have yet to be discovered [2]. Therefore, the intake of both provitamin A carotenoids and C8-keto carotenoids might effectively ameliorate inflammatory responses.

The plasma carotenoid concentration was reported to be about hundreds nM in a healthy human [26,27]. Nakagawa et al. reported that the plasma β-carotene concentration in Japanese women was about 1.6 µM [28]. Hartmann et al. showed that the plasma zeaxanthin concentration increased to 0.9 µM after 42-day ingestion of 10 mg per day of zeaxanthin [29]. Landrum et al. showed that daily ingestion of 30 mg of lutein for 140 days resulted in constitutively higher plasma levels of lutein (about 1.5 µM) [30]. Østerlie et al. reported that, in healthy men, the plasma astaxanthin concentration reached 2.2 µM at 6.7 h after administration of 100 mg of astaxanthin [31]. Based on these reports, it can be considered that the treatment concentration (1.0 µM) in this study is not too high to be reached in vivo. For the other carotenoids (echinenone, alloxanthin, siphonaxanthin, and fucoxanthin), there is no evidence that the plasma concentration can reach 1.0 µM. However, using mice and differentiated Caco-2 cells, a well-investigated model of human intestinal epithelium, we previously showed that siphonaxanthin can be absorbed from the intestine to the same extent as β-carotene and lutein [19]. Comparative studies on the intestinal absorption of carotenoids would also be important for further understanding of the anti-inflammatory activities of carotenoids.

β-Carotene 15,15′-oxygenase (BCO1) is an enzyme that generates retinal by cleaving provitamin A carotenoids (i.e., carotenoids with an unsubstituted β-ring) [32]. In humans, mRNA of BCO1 is expressed mainly in the small intestine, liver, and kidney [33]; however, to the best of our knowledge, there are no reports of BCO1 expression in monocytes. Using the CellExpress tool [34], we searched BCO1 expression levels in the Sanger Cell Line Project dataset and found that the expression level in THP-1 cells would be almost the same level as in LoVo colon cancer cells, which were demonstrated to express functional BCO1 [35]. Therefore, we considered that THP1-Dual cells possess BCO1-like activity, which plays a key role in mediating the anti-inflammatory effects of the provitamin A carotenoids. Oxidation at the C4 position of retinoids is a catabolic reaction. Cytochrome P450 26 is one of the enzymes that mediates this catabolism [36] and is reported to be induced by retinoic acid in THP-1 cells [37]. Therefore, not only echinenone but also β-carotene increased 4-oxo-retinal levels in the cells. Since all four retinoids in this study showed similar effects, we could not identify which retinoid was most important for the inhibition of the TNF-α-NF-κB axis and promotion of the PMA-IRF axis. However, for the first time, this study demonstrated that these biological activities are mediated by retinoids. The results of OPLS-DA indicate that at least these two biological activities are characteristic of the anti-inflammatory profiles of β-carotene and echinenone; however, these findings do not indicate that all of the effects of these two carotenoids are mediated by retinoids. In addition, we, in this study, demonstrated the increase in intracellular retinoid levels in β-carotene- and echinenone-treated cells but could not show whether β-carotene and echinenone served as a substrate of BCO1. Further studies are required to reveal the molecular mechanisms underlying the anti-inflammatory effects of each carotenoid.

Upon binding with LPS, TLR4 translocates into endosomes via lipid rafts, and endosomal TLR4 initiates the signaling pathway for IRF activation [38]. Hence, the molecular mechanism underlying the suppression of the LPS-IRF axis by C8-keto carotenoids might involve, in part, the inhibition of the lipid raft translocation of TLR4, as we previously reported [10]. In THP1-Dual cells, β-carotene did not accumulate but was metabolized to retinoids, suggesting that the non-polar carotenoid (β-carotene) did not mediate anti-inflammatory effects via modulation of lipid rafts. This result explains why the inhibition of the LPS-IRF axis was characteristic of C8-keto carotenoids.

The detailed signaling pathway of the P3C-IRF axis, the characteristic target of C8-keto carotenoids, is still controversial; however, recently, it has been reported that P3C induces IRF activation in THP1-Dual monocytes [39]. Thus, we considered that P3C itself, rather than any possible contaminants, induced IRF activation under our experimental conditions. Although its biological significance is not yet clear, TLR1/2 is known to translocate into lipid rafts upon binding with P3C [40]. Considering that the P3C-IRF axis and LPS-IRF axis are characteristic targets of the anti-inflammatory effects of C8-keto carotenoids, the modulation of receptor translocation into lipid rafts by C8-keto carotenoids may be involved in the inhibition of the P3C-IRF axis.

According to the Carotenoid Database [2], only 40 carotenoids with a carbonyl group at the C8 or C8′ position have been identified so far, and only some of them are found in foods such as seaweed and shellfish. However, it has been reported that Japanese women consume approximately 1 mg of the C8-keto carotenoid fucoxanthin per day, which is about 10% of their total carotenoid intake [41]. The anti-inflammatory potential of C8-keto carotenoids demonstrated in this study should be further investigated in future epidemiological studies.

Not only multivariate analyses but also each anti-inflammatory evaluation provided additional information. In the case of the IFN-γ-IRF axis, zeaxanthin showed more potent inhibition than astaxanthin (*p* = 0.01, Tukey-Kramer test), which has two additional keto groups at the C4 and C4′ positions. Echinenone also has an additional C4-keto group compared to β-carotene, but both are metabolized to retinoids in THP1-Dual cells. Taken together, these results suggest that the C4-keto group is an important carotenoid structure for enhancing the IFN-γ-mediated activation of IRFs. Contrary to the expectation from previous reports [11], alloxanthin, a C7,8-acetylenic carotenoid, did not suppress LPS-induced inflammatory responses, probably because the concentration used for the treatment was low. Compared to other inflammatory responses, the IFN-γ-IRF axis seemed to be more sensitive to carotenoids. In addition, this study is the first to report several anti-inflammatory effects, such as those of echinenone and siphonoxanthin. The next challenge is to confirm these findings by evaluating proinflammatory protein expression levels in primary monocytes.

In this study, we investigated the anti-inflammatory effects of eight carotenoids using nine different inflammatory inducers and by evaluating three different inflammatory responses. However, other carotenoids, inflammatory stimuli, and inflammatory responses need to be studied. Moreover, our research method can be applied to other food ingredients, natural compounds, and drug candidates. Expanding this research would provide in-depth information about anti-inflammatory compounds and their characteristics.

## Figures and Tables

**Figure 1 nutrients-13-03699-f001:**
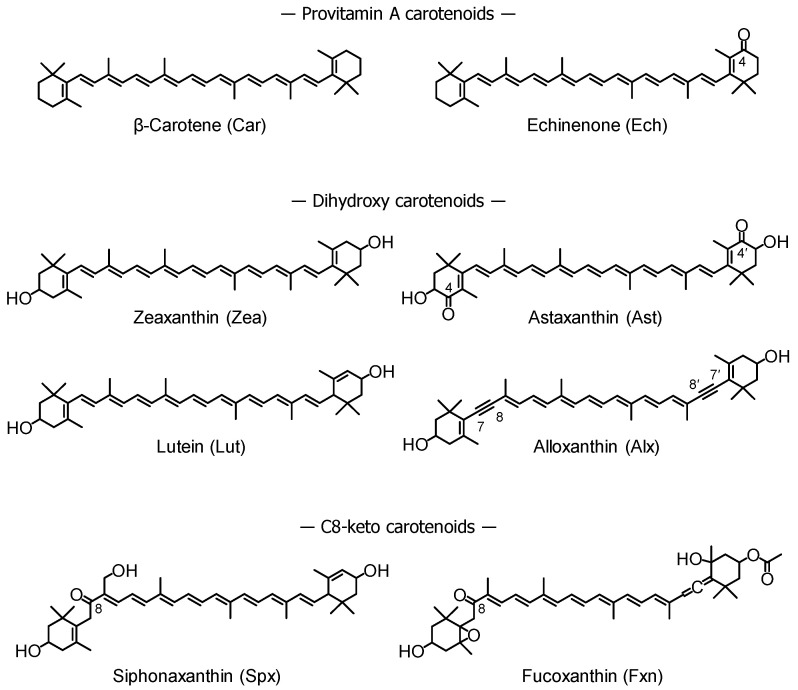
Chemical structures of the tested carotenoids. The three letters in parentheses are the corresponding abbreviations used in subsequent figures.

**Figure 2 nutrients-13-03699-f002:**
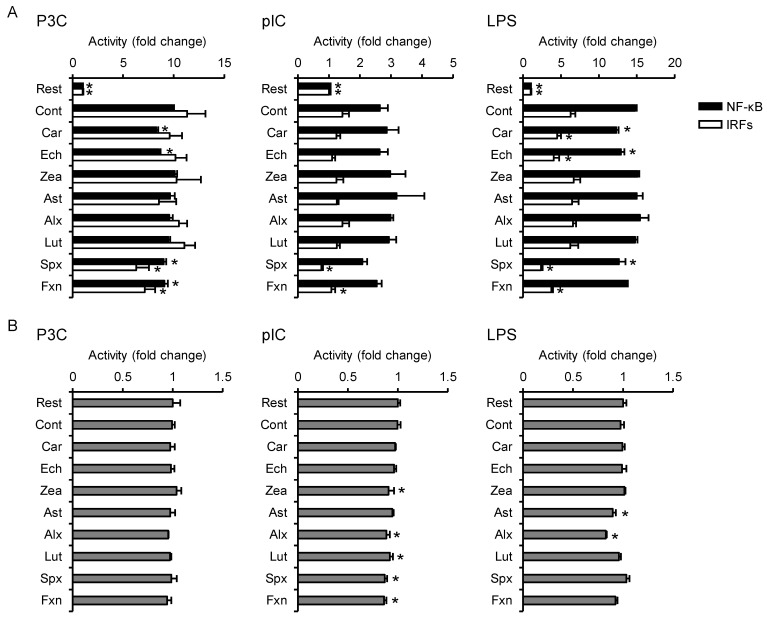
Effects of carotenoids on the activity of NF-κB, IRFs, and MTT reductase in TLR ligand-stimulated THP1-Dual cells. THP1-Dual cells were incubated with each carotenoid, followed by stimulation with the indicated TLR ligand. Carotenoids were dispersed in the culture medium using DMSO (maximum final concentration, 0.1%) as a vehicle. The resting cell and control groups were incubated with the DMSO-containing medium for 24 h, followed by no stimulation or stimulation with ligand, respectively. After stimulation, SEAP and luciferase activities in the supernatant (**A**) and MTT reductase activity in the cells (**B**) were measured (*n* = 4). Values represent mean ± SD. Significant differences were detected by one-way ANOVA in P3C-NF-κB (F(9,30) = 422, *p* < 0.05), P3C-IRF (F(9,30) = 20.6, *p* < 0.05), pIC-NF-κB (F(9,30) = 11.3, *p* < 0.05), pIC-IRF (F(9,30) = 8.17, *p* < 0.05), pIC-MTT reductase (F(9,30) = 12.2, *p* < 0.05), LPS-NF-κB (F(9,30) = 224, *p* < 0.05), LPS-IRF (F(9,30) = 40.9, *p* < 0.05), and LPS-MTT reductase (F(9,30) = 20.7, *p* < 0.05). The asterisks * indicate significant differences from the control groups (*p* < 0.05, Tukey-Kramer test). No significant difference was found by one-way ANOVA in P3C-MTT reductase (F(9,30) = 1.38, *p* = 0.239). Rest, resting cells; Cont, control; Car, β-carotene; Ech, echinenone; Zea, zeaxanthin; Ast, astaxanthin; Alx, alloxanthin; Lut, lutein; Spx, siphonaxanthin; Fxn, fucoxanthin.

**Figure 3 nutrients-13-03699-f003:**
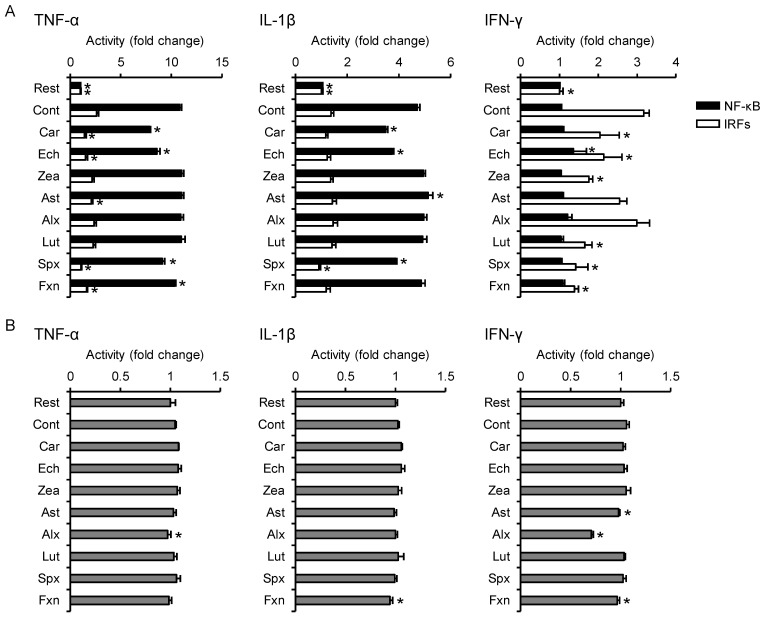
Effects of carotenoids on the activity of NF-κB, IRFs, and MTT reductase in cytokine-stimulated THP1-Dual cells. THP1-Dual cells were incubated with each carotenoid, followed by stimulation with the indicated cytokines. Carotenoids were dispersed in culture medium using DMSO (maximum final concentration, 0.1%) as a vehicle. The resting cell and control groups were incubated with the DMSO-containing medium for 24 h, followed by no stimulation or stimulation with cytokine, respectively. After stimulation, SEAP and luciferase activities in the supernatant (**A**) and MTT reductase activity in the cells (**B**) were measured (*n* = 4). Values represent mean ± SD. Significant differences were detected by one-way ANOVA in TNF-α-NF-κB (F(9,30) = 709, *p* < 0.05), TNF-α-IRF (F(9,30) = 39.4, *p* < 0.05), TNF-α-MTT reductase (F(9,30) = 6.40, *p* < 0.05), IL-1β-NF-κB (F(9,30) = 501, *p* < 0.05), IL-1β-IRF (F(9,30) = 9.62, *p* < 0.05), IL-1β-MTT reductase (F(9,30) = 5.80, *p* < 0.05), IFN-γ-NF-κB (F(9,30) = 3.39, *p* < 0.05), IFN-γ-IRF (F(9,30) = 26.0, *p* < 0.05), and IFN-γ-MTT reductase (F(9,30) = 57.8, *p* < 0.05). The asterisks * indicate significant differences from the control group (*p* < 0.05, Tukey-Kramer test). Rest, resting cells; Cont, control; Car, β-carotene; Ech, echinenone; Zea, zeaxanthin; Ast, astaxanthin; Alx, alloxanthin; Lut, lutein; Spx, siphonaxanthin; Fxn, fucoxanthin.

**Figure 4 nutrients-13-03699-f004:**
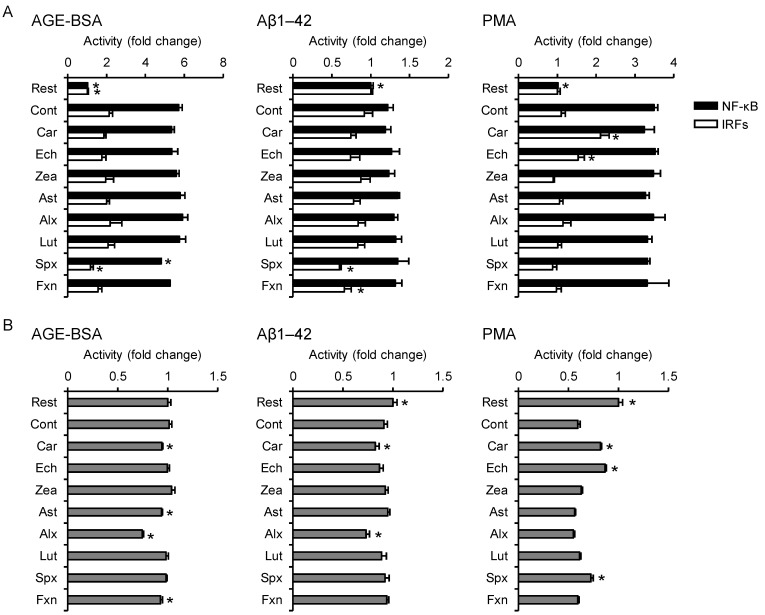
Effects of carotenoids on the activity of NF-κB, IRFs, and MTT reductase in AGE-BSA-, Aβ1–42-, or PMA-stimulated THP1-Dual cells. THP1-Dual cells were incubated with each carotenoid, followed by stimulation with the indicated agent. Carotenoids were dispersed in culture medium using DMSO (maximum final concentration, 0.1%) as a vehicle. The resting cell and control groups were incubated for 24 h, followed by no stimulation or stimulation with inflammatory inducer, respectively. After stimulation, SEAP and luciferase activities in the supernatant (**A**) and MTT reductase activity in the cells (**B**) were measured (*n* = 4). Values represent mean ± SD. Significant differences were detected by one-way ANOVA in AGE-BSA-NF-κB (F(9,30) = 222, *p* < 0.05), AGE-BSA-IRF (F(9,30) = 8.23, *p* < 0.05), AGE-BSA-MTT reductase (F(9,30) = 57.4, *p* < 0.05), Aβ1–42-NF-κB (F(9,30) = 7.21, *p* < 0.05), Aβ1–42-IRF (F(9,30) = 7.59, *p* < 0.05), Aβ1–42-MTT reductase (F(9,30) = 18.5, *p* < 0.05), PMA-NF-κB (F(9,30) = 41.4, *p* < 0.05), PMA-IRF (F(9,30) = 32.5, *p* < 0.05), and PMA-MTT reductase (F(9,30) = 234, *p* < 0.05). The asterisks * indicate significant differences from the control group (*p* < 0.05, Tukey-Kramer test). Rest, resting cells; Cont, control; Car, β-carotene; Ech, echinenone; Zea, zeaxanthin; Ast, astaxanthin; Alx, alloxanthin; Lut, lutein; Spx, siphonaxanthin; Fxn, fucoxanthin.

**Figure 5 nutrients-13-03699-f005:**
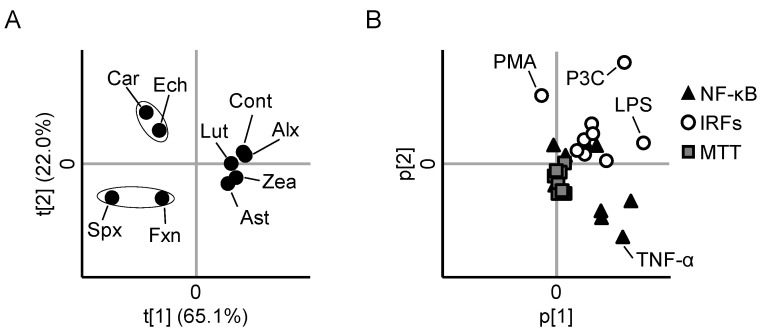
Principal component analysis of anti-inflammatory activity of each carotenoid. Score plot (**A**) and loading plot (**B**) of principal component analysis of activities of NF-κB, IRFs, and MTT reductase in stimulated THP1-Dual cells.

**Figure 6 nutrients-13-03699-f006:**
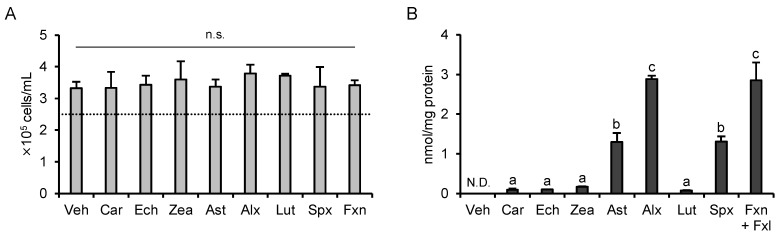
Cell density and cellular accumulation of each carotenoid after treatment. THP1-Dual cells were incubated with each carotenoid (1.0 µM) for 24 h. Carotenoids were dispersed in culture medium using DMSO (maximum final concentration, 0.1%) as a vehicle. The vehicle group received only DMSO. After treatment, cell density (**A**) and cellular accumulation of each carotenoid (**B**) were measured (*n* = 3). Values represent mean ± SD. No significant difference was detected in cell density (F(8,18) = 0.649, *p* = 0.728), whereas in cellular carotenoid accumulation, a significant difference was found by one-way ANOVA (F(8,18) = 136, *p* < 0.05). The different characters represent significant differences among treatments, and n.s. indicates no significance (*p* < 0.05, Tukey-Kramer test). Veh, vehicle; Car, β-carotene; Ech, echinenone; Zea, zeaxanthin; Ast, astaxanthin; Alx, alloxanthin; Lut, lutein; Spx, siphonaxanthin; Fxn, fucoxanthin; Fxl, fucoxanthinol.

**Figure 7 nutrients-13-03699-f007:**
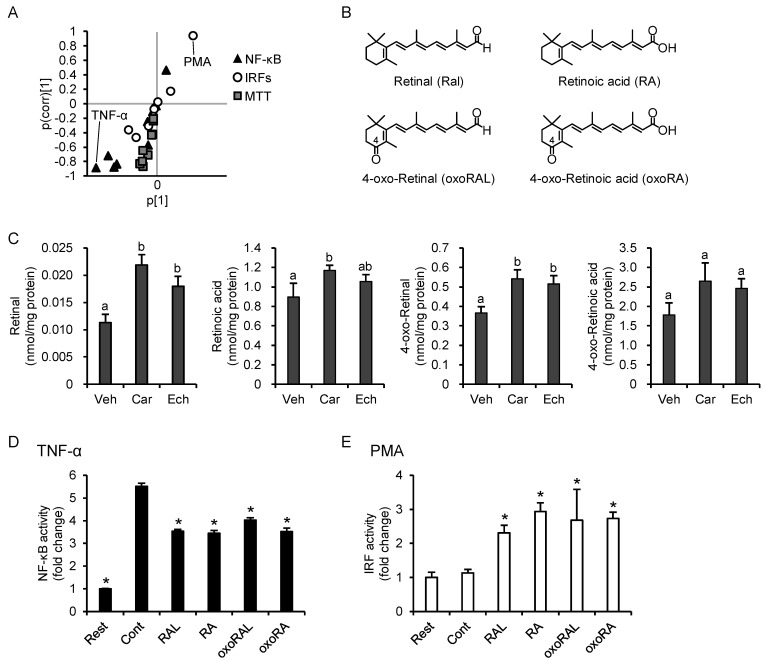
Characteristics and anti-inflammatory profiles of provitamin A carotenoids. (**A**) S-plot of orthogonal partial least squares discriminate analysis of NF-κB, IRFs, and MTT reductase in β-carotene- or echinenone-treated cells and those in the other carotenoid-treated cells. (**B**) Chemical structures of the tested retinoids. The letters in parentheses are abbreviations used in subsequent panels. (**C**) THP1-Dual cells were incubated with carotenoids (1.0 µM), which were dispersed in the culture medium using DMSO (maximum final concentration, 0.1%) as a vehicle. The vehicle group received DMSO only. After 24 h of incubation, cellular retinoid levels were quantified using LC-MS/MS (*n* = 3). Values represent the mean ± SD. (**D**,**E**) THP1-Dual cells were incubated with each retinoid at 1.0 µM for 24 h, followed by stimulation with TNF-α (**D**) or PMA (**E**). Retinoids were dispersed in the culture medium using DMSO (maximum final concentration, 0.1%) as a vehicle. The resting cell and control groups were incubated with the DMSO-containing medium for 24 h, followed by no stimulation or stimulation, respectively. After 24-h stimulation, SEAP (**D**) and luciferase activities in the supernatant were measured (*n* = 4). Values represent the mean ± SD. Significant differences were detected by one-way ANOVA in retinal (F(2,6) = 27.8, *p* < 0.05), retinoic acid (F(2,6) = 5.98, *p* < 0.05), 4-oxo-retinal (F(2,6) = 16.1, *p* < 0.05), TNF-α-NF-κB (F(5,18) = 650, *p* < 0.05), and PMA-IRF (F(5,18) = 17.7, *p* < 0.05). The different characters over the bar represent significant differences among treatments, and the asterisks * indicate significant difference from the control group (*p* < 0.05, Tukey-Kramer test). No significant difference was detected in one-way ANOVA in 4-oxo-retinoic acid (F(2,6) = 4.84, *p* = 0.056). Veh, vehicle; Car, β-carotene; Ech, echinenone; Rest, resting cells; Cont, control; RAL, retinal; RA, retinoic acid; oxoRAL, 4-oxo-retinal; oxoRA, 4-oxo-retinoic acid.

**Figure 8 nutrients-13-03699-f008:**
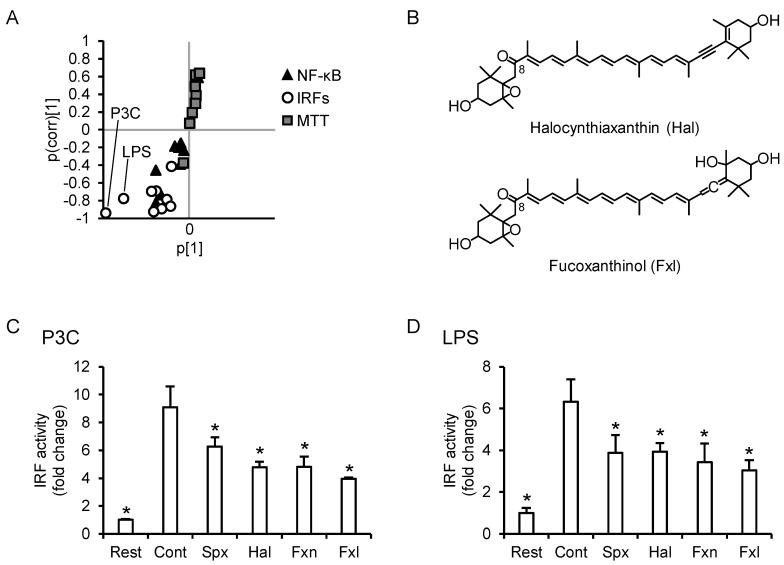
Characteristics and anti-inflammatory profiles of C8-keto carotenoids. (**A**) S-plot of orthogonal partial least squares discriminate analysis of NF-κB, IRFs, and MTT reductase in siphonaxanthin- or fucoxanthin-treated cells and those in the other carotenoid-treated cells. (**B**) Chemical structures of additionally tested C8-keto carotenoids. The three letters in parentheses are abbreviations used in subsequent panels. (**C**,**D**) THP1-Dual cells were incubated with each carotenoid (1.0 µM) for 24 h, followed by stimulation with P3C (**C**) or LPS (**D**). Carotenoids were dispersed using DMSO (maximum final concentration, 0.1%) as a vehicle, and resting cell and control groups were incubated with the DMSO-containing medium for 24 h, followed by no stimulation or stimulation with ligand, respectively. After stimulation, luciferase activity in the supernatant was measured (*n* = 4). Values represent the mean ± SD. Significant differences were detected by one-way ANOVA in P3C-IRF (F(5,18) = 49.2, *p* < 0.05) and LPS-IRF (F(5,18) = 22.5, *p* < 0.05). The asterisks * indicate significant difference from the control group (*p* < 0.05, Tukey-Kramer test). Rest, resting cells; Cont, control; Spx, siphonaxanthin; Hal, halocynthiaxanthin; Fxn, fucoxanthin; Fxl, fucoxanthinol.

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
