# Peer review of "Multivariate Analysis Reveals That Unsubstituted β-Ring and C8-Keto Structures Are Important Factors for Anti-Inflammatory Activity of Carotenoids"

_nutrients, 2021, doi:10.3390/nu13113699_

Round 1

Reviewer 1 Report

In their manuscript Manabe et al. present a systematic and comprehensive analyses of the preventive activities of 8 different carotenoids on activation of the NF-kB and IRF pathways. For such studies authors used stable human monocyte line with reporter systems for monitoring of both pathways. The use of different pro-inflammatory stimuli pertinent for investigation of different mechanisms of NF-kB and NRF activation, makes this study of general interest and a prototype of resource article. This is well designed and clearly described study with interesting results, which open new avenues for investigation. I have only minor remarks:

  • As indicated by authors many carotenoids are metabolized mainly in the intestine or liver, so it is not clear to which extent monocytes can be exposed to some of the tested carotenoids in physiological or pathological conditions in human. The paper would benefit from adding or discussing information on plasma levels of such carotenoids and/or their metabolites in human.
  • similarly, in line 75 authors indicate that carotenoids were used at physiologically relevant concentrations while in the text only refs 16 and 17 are cited concerning mainly b-carotene. This statement should be better documented.
  • According to materials and methods, I understand that each carotenoid was added in DMSO at 24hrs prior to pro-inflammatory stimuli, which lasted for the following 24hrs in presence of carotenoids. If this is correct, that is fine, but if not, please add a drawing of an experimental design
  • Line 365: authors indicate that THP1 cells possess BCO1 activity. Please, either moderate the statement to “BCO1-like” or show presence of BCO1 in TPH1 cells.
  • In figure 7C, authors elude to significant differences in the quantity of retinoids after Car or Ech treatment, but the meaning of symbols “a” and “b” is not reported. Please report statistics for this graph.
  • Retinoids are reported as nmol/mg of protein, please indicate in MMs how protein level was evaluated
  • Please, show in supplementary data examples of detection of specific retinoids by LC-MS
  • Indicate number of samples “n” used for each type of measure
  • Authors report using one-way ANOVA, but no results of such analyses are reported - please provide F values, p values etc for comparisons where ANOVA was used

Reviewer 2 Report

KEYWORDS: I strongly suggest authors to introduce more keywords. The usefulness of keywords is to make the article both more and more easily searchable visible after its publication through commonly used search engines..

Introduction: The introduction is interesting, but in my opinion it does not fully cover the topic. Moreover, out of 13 cited items, some are older than 10 years. The authors refer to  some very old literature (item 3, 5, 12). Can those items be replaced with newer one?

It should be noted that the authors described Materials and Methods very thoroughly, as well as the description and graphic representation of the results were presented very well. I believe that the work presented for review is of a high technical level. I am asking for a deeper description, taking into account my suggestions above, with post new items.

Reviewer 3 Report

The manuscript by Manabe et al, delves into potential anti-inflammatory activities of various carotenoids. This is a subject that has received a lot of attention due to the the presence of these bioactive nutrients in our diet, however, there is a lack of consensus and the mechanisms involved remain unclear. The authors use a large set of proinflammatory stimulators to treat NF-κB reporter cells concurrently exposed to a single-dose of various carotenoids. The data is then collected at a single time point. The results are potentially interesting but preliminary due to lack of dose-response studies, time-course and validation of the carotenoid treatment. More importantly, the studies do not provide significant new insight as to the possible anti-inflammatory mechanisms of these carotenoids.

1) The authors use preparations of carotenoids and retinoids from both natural sources and commercial suppliers without indicating how the purity and concentrations were validated. The authors refer the reader to Ref 14 and 15 in relation to carotenoid purification but neither of these references provide the necessary details. 
2) It would have been preferable to use a dose-response combined with validation of the cellular levels of carotenoids (see below).  The authors employ a single dose of 1uM carotenoid, which is referred to as a physiologically relevant concentration, though it is not clear what a physiological level means in the context of carotenoids not commonly found in diet, like alloxanthin for example. Though, there are examples of carotenoids reaching micromolar concentrations in some settings, there is insufficient evidence that this concentration represents an attainable level in vivo for many of the carotenoids discussed here. Additionally, a dose-response study using several concentrations of carotenoids would have provide much more statistical power and more confidence that the effects observed are related to treatment.
3) It is clear that the carotenoids studied are absorbed and processed at very different rates. This makes the comparison of their cellular effects difficult. It is also difficult to distinguish the effects of uncleaved and apocarotenoid metabolites without further analysis. 
4) It would be important to determine if BCO1 and BCO2 are expressed in the THP1-Dual cell line. The authors presume that the lower (in comparison to other carotenoids) final steady state level of β-carotene and echinenone is a result of metabolism to retinoids, but it would have been important to establish the levels of parent carotenoid and product metabolites via a time course.
5) The anti-inflammatory mechanism of carotenoids studied here is insufficiently explored. Much more could have been done, for example, does antagonism of RAR abrogate the anti-inflammatory effects of  β-carotene and echinenone? 

Round 2

Reviewer 3 Report

The authors have adequately addressed my concerns given the scope of their study and the available timeframe by revising the manuscript and providing helpful clarifications.